# Development of Positron Emission Tomography Radiotracers for Imaging α-Synuclein Aggregates

**DOI:** 10.3390/cells14120907

**Published:** 2025-06-16

**Authors:** Xiaodi Guo, Jie Xiang, Keqiang Ye, Zhentao Zhang

**Affiliations:** 1Department of Neurology, Renmin Hospital of Wuhan University, Wuhan 430060, China; guoxiaodii@163.com; 2Department of Neurobiology, Fourth Military Medical University, Xi’an 710032, China; xj890119@fmmu.edu.cn; 3Faculty of Life and Health Sciences, Shenzhen University of Advanced Technology (SUAT), Shenzhen 518055, China; 4TaiKang Center for Life and Medical Sciences, Wuhan University, Wuhan 430000, China

**Keywords:** synucleinopathy, Parkinson’s disease, tracer, high-throughput screening

## Abstract

Neurodegenerative diseases (NDDs) that are characterized by the accumulation of alpha-synuclein (α-syn) aggregates in both neurons and the non-neuronal cells of the brain are called synucleinopathies. The most common synucleinopathies includes Parkinson’s disease (PD), Parkinson’s disease dementia (PDD), multiple system atrophy (MSA), and dementia with Lewy bodies (DLB). Significant progress has been made in the development of positron emission tomography (PET) radiotracers for synucleinopathies, yielding several α-syn tracers that have entered clinical studies. However, selective α-syn imaging still faces inherent challenges. This review provides a comprehensive overview of the progress in α-syn PET radiotracers from three angles: Alzheimer’s disease (AD)-derived scaffolds, representative compound scaffolds and analogs, and the identification of α-syn tracers through high-throughput screening (HTS). We discuss the characteristics, advantages, and limitations of the tracers for preclinical and clinical application. Finally, future directions in the development of radioligands for proteinopathies are discussed. There is no clinical available PET radiotracer for imaging α-syn aggregates, but these advances have laid a key foundation for non-invasive α-syn imaging and early diagnosis of synucleinopathies.

## 1. Introduction

The aberrant aggregation of characteristic proteins has been recognized as a key molecular mechanism in many neurodegenerative diseases (NDDs) [1]. NDDs characterized by the accumulation of alpha-synuclein (α-syn) in neurons and non-neuronal cells in the brain are called synucleinopathies, including Parkinson’s disease (PD), Parkinson’s disease dementia (PDD), dementia with Lewy bodies (DLB), multiple system atrophy (MSA) and pure autonomic failure (PAF) [2,3]. The prevalence of synucleinopathies is positively correlated with age [4]. PD is the most common synucleinopathy, affecting approximately 1–2% of the population over 65 years old worldwide. PD has become one of the fastest-growing neurological diseases in the world. The corresponding global burden has more than doubled compared with that of the previous generation [5,6,7,8,9]. According to a previous study, the number of PD patients in the five most populous countries in Western Europe and the ten most populous countries in the world is expected to double by 2030, reaching between 8.7 and 9.3 million [8]. In addition, a previous study revealed that the risk of PD is greater in men [10], reaching a peak in the age groups of 85–89 years for men and 90–94 years for women [11]. This poses a great economic burden and public health challenge to modern society [8,12]. With the establishment of public health strategies for the prevention of cardiovascular disease and certain cancers, chronic NDDs such as PD are gaining increased attention [13].

PD, PDD, and DLB are characterized by α-syn in neurons and intracytoplasmic Lewy body (LB) formation, with α-syn deposition occurring primarily in the substantia nigra in PD patients and mainly in cortical areas in DLB patients [14,15]. MSA is characterized by the deposition of α-syn in glial cytoplasmic inclusions (GCIs) [16,17]. Therefore, α-syn is not only an important molecule in the research of PD pathology but also a valuable imaging target. However, the early diagnosis and differential diagnosis of this disease are very difficult. Currently, the diagnosis of PD is mainly based on history, clinical examination (movement difficulties such as tremor, stiffness, bradykinesia, and postural instability), and the response to pharmacological treatment [18,19]. At the same time, trauma- or drug-induced Parkinsonism must be excluded [20]. Furthermore, clinicopathological studies have shown that more than a quarter of patients diagnosed with idiopathic PD during their lifetime are misdiagnosed. A significant proportion of patients clinically diagnosed with PD do not meet histopathological PD criteria at autopsy. These patients mainly suffer from “Parkinsonism” due to other causes (i.e., atypical Parkinsonism, Alzheimer-type pathology, or vascular changes) [20,21,22]. This poses a problem because the definitive signs and symptoms of synucleinopathies (such as movement disorders in PD and cognitive impairment in DLB) often manifest in the late stages of disease, when pathological aggregates of α-syn are widespread [23,24]. In addition, the symptoms of PD overlap with those of essential tremor and “atypical” Parkinsonism, such as DLB, MSA, and progressive supranuclear palsy (PSP), which occur more frequently in old patients, thus confounding the diagnosis of PD in elderly patients [24,25].

It should be noted that neuropathological evidence demonstrates a significant co-existence of pathological proteins between PD and AD (copathology) [26,27,28,29]. Notably, postmortem studies reveal that over 50% of PD cases exhibit concomitant AD-type pathology [30,31,32], which clinically correlates with accelerated disease progression and the earlier onset of dementia in PD patients [33]. Similarly, α-syn inclusions have been identified in the brains of AD patients [26,34]. The co-existence of these pathologies has been associated with worse cognitive decline and increased mortality in patients with the disease [35]. Recent evidence suggests that amyloid-β (Aβ), tau, and α-syn interact and influence each other through multiple molecular mechanisms [36,37,38]. α-Syn/tau can interact directly, to form aggregates for in vitro seeding [39,40], or indirectly, through kinases such as glycogen synthase kinase-3β (GSK-3β), leucine-rich repeat kinase 2 (LRRK2), and cyclin-dependent kinase 5 (CDK5) [41,42,43,44]. In vitro studies show co-aggregation of Aβ and α-Syn and their direct interaction through the C-terminus of Aβ [45]. Aβ can also play an indirect role by inducing the phosphorylation of α-syn by several kinases, such as Cdk5, casein kinase 2 (CK2), and polo-like kinase 2 (PLK2) [44,46]. The co-existence of multiple protein aggregates in a single disease or multiple diseases is thought to be due to cross-seeding between these proteins [47], which makes the diagnosis and treatment of the diseases more complicated. Therefore, the frequent presence of mixed pathologies makes it necessary to confirm the pathological changes in brain tissue [23].

Synucleinopathies present protracted premotor phases, such as olfactory changes [48] and the development of rapid eye movement (REM) sleep behavior disorders [49,50,51]. Clinical Parkinsonism develops when PD patients have lost at least 50% of their substantia nigra cells, at which time the loss of dopaminergic nerve endings in the posterior putamen may be more severe [52,53]. Braak et al. reported that the deposition of α-syn in LBs and Lewy neurites (LNs) can be divided into six consecutive stages, with sequential accumulation and expansion occurring [54]. Most positron emission tomography (PET)/single-photon emission computed tomography (SPECT) tracers developed on the basis of the function of the dopaminergic system only show abnormal dopaminergic imaging via PET or SPECT when many dopaminergic neurons in the substantia nigra pars compacta (SNpc) are lost [55]. These imaging techniques help differentiate PD with motor symptoms from disorders that do not involve neuronal loss in the SNpc, such as essential tremor. Studies have shown that, in some cases, imaging evidence of cardiac denervation precedes imaging findings of nigrostriatal dopaminergic damage [56,57]. Dopamine imaging alone is not sufficient to definitively diagnose PD because it cannot effectively distinguish PD from other Parkinsonisms involving substantia nigra degeneration. Therefore, detection methods based on α-syn imaging rather than dopaminergic changes will be the key to achieving the earliest and most accurate detection of premotor PD [23].

Various studies have highlighted the critical role of imaging techniques for patient inclusion in clinical trials, especially for PD [58]. The advent of non-invasive molecular imaging methods such as magnetic resonance imaging (MRI), SPECT, and PET has made it possible to distinguish PD from atypical Parkinsonisms in living patients [59]. MRI enables the visualization of slow oscillatory changes in cerebral venous oxygenation and the detection of structural damage, especially changes in the structure of substantia nigra, striatum, brainstem, and cerebellum in parkinsonian syndromes [60]. SPECT, while highly sensitive for detecting nigrostriatal degeneration in PD, suffers from limited quantitative reliability and requires integration with other techniques (e.g., transcranial sonography, olfactory testing) to enhance diagnostic accuracy [61]. Critically, neither is fully specific to PD, and the occurrence of early-stage abnormalities during disease progression remains unknown. Furthermore, these approaches lack the ability to directly detect the abnormal aggregation of α-syn [62]. PET imaging, a potential imaging technique for the non-invasive direct detection of α-syn aggregation, has been successfully applied to visualize Aβ and tau proteins in AD patients but remains in the early stages of development for α-syn detection [59]. The identification of small-molecule PET radiotracers with high affinity and specificity for fibrillated α-syn will enable the precise quantification of α-syn aggregation in the brain. Once suitable tracers are established, PET molecular imaging is expected to provide critical insights into disease pathogenesis, facilitate the localization and quantification of drug targets at the molecular level, monitor treatment responses, and visualize disease progression. This advancement will not only improve the accuracy and timeliness of clinical diagnosis but also serve as a crucial tool for the development of α-syn-targeted therapeutics. In this review, we aim to comprehensively summarize in detail the latest advancements and ongoing challenges in the field of α-syn PET tracers and explore potential future research directions.

## 2. Challenges in Developing α-Syn PET Tracers

The development of α-syn PET tracers has focused primarily on the use of highly specific radioactive molecular markers for α-syn to visualize its abnormal aggregation in vivo. These tracers are expected to be used for the early diagnosis, progression monitoring, and efficacy evaluation of α-synucleinopathies. In 2011, the Michael J. Fox Foundation formed a consortium of researchers aimed at developing α-syn PET radiotracers. They adopted small-molecule pharmacochemistry methods, starting with the screening of compounds and identifying novel lead compounds that selectively bind to α-syn. After continued optimization, the most promising compounds were radiolabeled for further development [63]. Compared with radiotracers for other amyloid proteins (such as Aβ or tau), the development of PET tracers for α-syn aggregates faces greater challenges [64,65].

Currently, no radiotracers have been approved for human α-syn imaging due to several challenges [63,65,66]. First, although α-syn is abundant in the central nervous system, accounting for 1% of the total protein content in the brain cytosol, it primarily exists in oligomeric forms [67] and has a relatively weak binding capacity for small molecules. In diseased brains, the quantity of α-syn aggregates is relatively low, ranging from only 1/10 to 1/50 of that of Aβ or tau aggregates, and the concentration in the brainstem and subcortical regions is only 50–200 nM in late-stage cases [68]. These aggregates are small in size and widely distributed, making their detection very difficult [68]. Second, α-syn often co-localizes with Aβ [31,69,70,71] and accumulates with various proteins (e.g., synphilin-1 and parkin) within cells [72]. Distinguishing these aggregates requires tracers with high specificity and affinity, as well as sufficient radioactive signals, to ensure imaging quality for the accurate detection of α-syn. Third, candidates should exhibit high blood–brain barrier (BBB) permeability, with a standardized uptake value (SUV) greater than 1.0 in the brain within minutes (5–10 min) after intravenous injection [73]. It is recommended that amyloid-targeting tracers achieve brain uptake ≥0.4% of the injected dose per gram (% ID/g) in rats or ≥4.0% ID/g in mice [73]. The majority of α-syn aggregates reside intracellularly, predominantly localized within neurons or glial cells. In addition to being able to pass through the BBB, tracers need to cross the cell membrane via active/passive transport [73,74]. However, for effective BBB penetration via passive diffusion, candidates should be relatively small (<700 Da) and exhibit moderate lipophilicity (logD_7.4_ < 3) to avoid nonspecific tracer retention and ensure rapid brain clearance [75]. Fourth, α-syn aggregates are structurally complex and undergo a variety of post-translational modifications [76,77], all of which should be detectable by ideal tracers. Modified forms of α-syn affect cytotoxicity, solubility, and the propensity to form oligomers and fibrils. This heterogeneity of α-syn makes it difficult to establish in vitro assays to optimize lead compounds [66]. The most common form of modification is phosphorylation. In the DLB brain, the vast majority of insoluble α-syn is phosphorylated at serine 129 [78]. Another common post-translational modification is ubiquitination, which may occur after phosphorylated synuclein is deposited in LBs and affects the solubility of the deposited α-syn [79]. Furthermore, when inflammation and oxidative stress levels are elevated in brain regions, α-syn can undergo oxidative modifications, such as the nitration of tyrosine residues and oxidation of methionine sulfonamide, both of which have been observed in postmortem PD brain slices [77]. Another key factor is the density of binding sites in the brains of PD patients. To ensure effective imaging with high-affinity ligands, there should be sufficient binding sites [80,81]. Interestingly, α-syn is also found in peripheral tissues such as the gastrointestinal tract, but its amount may not be sufficient for imaging [82,83]. In addition, attention should be given to the transformation of animal models into human applications. The initial in vivo evaluation of candidate brain imaging tracers is often performed in rodents, but BBB permeability and receptor richness vary significantly in different species [84,85]. Although these animal models cannot fully reproduce the pathological process of human disease, they are useful for evaluating in vivo stability, brain penetration, and the targeting of tracers. In addition, ideal α-syn tracers should have good clearance from the brain after nonspecific uptake to ensure a high signal-to-noise ratio, no P-glycoprotein substrate activity, no production of radioactive metabolites, and properties suitable for quantification and obtaining reproducible measurements, preferably by simple and non-invasive analysis [74]. Overall, ideal α-syn tracers should meet these criteria to accurately and comprehensively examine the distribution of pathological α-syn in vivo, a task that is globally challenging.

## 3. From AD-Radioactive Tracers and New Derivatives

Aβ, α-syn, and tau proteins form similar β-sheet structures when they aggregate [86,87,88], an important factor for PET tracers for these proteinopathies. Therefore, tracers that can interact with Aβ aggregates have the potential to bind to aggregated α-syn for the multiscale assessment of intracellular α-syn fibrils. Hence, several established Aβ PET probes have been evaluated against α-syn. Notable examples include the radiotracers [^11^C]-Pittsburgh compound B ([^11^C]-PIB), [^11^C]-2-[2-(2-dimethylaminothiazol-5-yl)ethenyl]-6-[2-(fluoro)ethoxy] benzoxazole ([^11^C]-BF227), 2-((1E,3E)-4-(6-([^11^C]methylamino)pyridin-3-yl)buta-1,3-dienyl) benzo[d]thiazol-6-ol ([^11^C]PBB3), and their structural analogs, which have been used in imaging experiments on α-syn aggregates. All tracers and new derivatives from AD and their chemical structures are listed in Table 1 and Figure 1, respectively.

### 3.1. PIB

[^3^H]-PIB, a thioflavin-T (ThT) derivative, is the gold standard for staining all types of amyloid proteins. [^3^H]-PIB has been shown to bind to recombinant α-syn fibrils with low affinity in vitro (K_d_ = 10.07 nM) and higher affinity for Aβ_1-42_ fibrils (K_d_ = 0.71 nM) [89]. However, in vitro studies have shown that PIB does not bind to pure postmortem DLB brain homogenates [89]. In amygdala sections from postmortem PD patients, LBs were not detected to bind to PIB by autoradiography (ARG) [90]. In DLB brains, Aβ plaques occupy 35 times the area of Lewy bodies, so [^11^C]-PIB for PET cannot distinguish between the two overlapping NDDs [89]. Additionally, in a [^3^H]PiB competition binding assay, 6-fluoro-2-(4-(piperidin-1-yl)styryl)benzo[*d*]thiazole (PFSB) (K_i_ = 25.4 ± 2.3 nM) and a less lipophilic analog, 4-(4-(2-(6-fluorobenzo[d]thiazol-2-yl)phenyl)morpholine (MFSB), which has a stronger affinity for α-syn (K_i_ = 10.3 ± 4.7 nM) and retains selectivity for Aβ [91], were tested. Both lead compounds were labeled with ^18^F, and an ARG assay revealed selective binding to α-syn in brain sections from MSA patients in vitro [91]. PET/MRI revealed that [^18^F]MFSB was able to cross the BBB (standardized uptake value (SUV) = 1.79 ± 0.02)) despite slow brain uptake and poor washout [91]. Researchers have noted that the pharmacokinetic properties of [^18^F]MFSB require further structural optimization in the future.

### 3.2. BF227

[^18^F]BF227 was initially developed for imaging amyloid plaques in AD [92]. Subsequent in vitro studies on the basis of its strong fluorescence properties revealed that [^18^F]BF227 was nonselective and had a high binding affinity for recombinant α-syn fibrils (K_d_ = 9.6 nM), but that this was still more than seven times lower than its affinity for Aβ_1-42_ fibrils (K_d_ = 1.31 nM) [93]. In vitro binding studies revealed that [^18^F]-BF227 failed to bind to pure LBD brain homogenates [93]. This yielded results similar to those of previous studies evaluating PiB binding to α-syn fibrils and pure LB brain homogenates. Furthermore, Levigoureux et al. evaluated the α-syn aggregate binding capacity of [^18^F]BF227 via in vitro ARG and microPET imaging in a transgenic mouse model (TgM83) expressing human A53T-mutated α-syn and α-syn knockout mice and reported no significant difference in [^18^F]BF227 binding levels in the brainstem, cerebellum, or thalamus between the transgenic and control mice, indicating that [^18^F]BF227 did not specifically bind to α-syn aggregates in TgM83 mice [94]. In a subsequent study, in vitro ARG of [^18^F]-BF227 in the brains of MSA patients did not support the idea that [^18^F]BF-227 is present at the same concentrations as GCIs which are typically observed in PET experiments [95]. In addition, a PET study of [^11^C]BF227 revealed that it was able to detect α-syn deposits in the living brains of patients with MSA, leading to the conclusion that [^11^C]BF227 is able to image α-syn aggregates in vivo but with low selectivity [96].

To improve the affinity and selectivity for α-syn fibrils and to verify the interaction with Aβ fibrils, Josephson designed a series of BF-227-like benzoxazoles by introducing various oxyethylene groups as well as hydrogen, iodine, and fluoride [97]. The introduction of oxyethylene groups had no effect on the affinity or selectivity of the compounds. The substitution of the fluoroethoxy group with iodine improved the selectivity, whereas substitution with fluorine or hydrogen reduced the affinity for α-syn fibrils more than five-fold [97]. However, the results concerning the binding affinity of BF-227 for α-syn fibrils differ from those of the initial report by Fodero-Tavoletti et al. [93], probably because of the different assay methods and fibril preparation methods used in vitro.

### 3.3. PBB3

[^11^C]-PBB3 is one of the first generation of tau radiotracers. [^11^C]-PBB3 labels a variety of α-syn lesions. Postmortem fluorescence and autoradiographic evidence confirm that PBB3 binds to α-syn in LB disease and MSA [98]. Studies with similar results suggest that [^11^C]-PBB3 binding is present in asymptomatic α-syn repeat carriers and MSA patients in the absence of tau pathology [99]. However, its binding affinity for α-syn aggregates has been shown to be insufficient for the sensitive PET detection of these lesions in living individuals [99]. Recently, [^3^H]-C05-01, an amino-pyridinyl-butenynyl-benzothiazole derivative, was developed and was shown to have a K_i_ of 3.5 nM for brain homogenates and 25 nM for α-syn fibrils [100]. ARG studies using fresh frozen human tissues and tissue microarrays (TMAs) revealed that [^3^H]C05-01 specifically bound to PD and MSA cases, but C05-01 specifically bound to Aβ and tau pathology in AD tissues, with relatively high nonspecific and off-target binding [100]. Therefore, its specificity for α-syn is limited.

There is also a PBB3 derivative with an (E)-hex-2-en-4-yne linker, the C05 series compound ((E)-1-fluoro-3-((2-(4-(6-(methylamino)pyridine-3-yl)but-1-en-3-yn-1-yl)benzo[d]thiazol-6-yl)oxy)propan-2-ol), a structure that results in the significantly increased binding of the ligand to α-syn relative to tau and Aβ fibrils [101]. Recently, Endo’s team developed a new small-molecule ligand, C05-05, and successfully visualized Lewy lesions in the midbrain, which is considered superior to C05-01 as an in vivo imaging agent [101]. In vivo optical and PET imaging in mouse and marmoset models demonstrated that C05-05 visualized the dynamic propagation of fibers along neural pathways, followed by the disruption of these structures [101]. In vitro ARG revealed that ^18^F-C05-05 binds with high affinity to α-syn pathology in DLB, MSA, and PD brain tissues [101]. Notably, this study demonstrated the utility of ^18^F-C05-05 for detecting α-syn deposits in the brains of PD patients and DLB patients via PET, demonstrating for the first time the visualization of α-syn pathology [101]. Overall, the study presents a new and impactful imaging technique, but it did not include PD and DLB patients with PET-detectable Aβ and tau pathology, so the specificity of the tracer remains to be investigated.

PET ligands for the detection of α-syn pathology have recently been applied to patients with MSA [102,103]. Higuchi and colleagues developed a small-molecule ligand for α-syn fibrils, ^18^F-SPAL-T-06, which is structurally similar to the C05 series for the PET imaging of synucleinopathies, and first achieved in vivo imaging in patients with MSA [102]. MSA patients presented increased ^18^F-SPAL-T-06 retention in the putamen and increased ^18^F-SPAL-T-06 accumulation in the pons and cerebellar white matter. Radioligand binding was also enhanced in the cerebellar pedicle of the MSA patients, especially in patients with predominant cerebellar ataxia (MSA-C) [102]. However, as this study included only four participants, further validations are needed. In another study on GCI imaging in MSA patients, Ruben Smith and colleagues used AC Immune’s proprietary Morphomer^®^ library to identify a brain-permeable small molecule ([^18^F]ACI-12589) with high affinity for α-syn aggregates and good selectivity for other potential brain pathologies (or copathologies) [103]. Encouragingly, [^18^F]ACI-12589 had a K_d_ value of 33.5 nM and excellent in vitro selectivity for α-syn over pathological Aβ, tau, and TDP-43 [103]. [^18^F]ACI-12589 showed different uptake patterns in brain regions in MSA-C and MSA with Parkinsonism predominant (MSA-P), with high PET signals of [^18^F]ACI-12589 observed in the cerebellar white matter and cerebellar midpedicle in both. In addition, MSA-C participants presented increased PET signal retention in cerebellar structures, whereas participants with MSA-P presented uptake in the lentiform nucleus, suggesting the involvement of the basal ganglia [103]. [^18^F]ACI-12589 regional preservation in the cerebellar white matter and cerebellar peduncles of MSA patients clearly distinguished MSA patients from controls and PD or DLB patients, which was consistent with previous reports [103,104]. [^18^F]ACI-12589 can bind to α-syn in AD and PSP tissues in vitro, so some tracer retention was also observed in non-MSA patients, and some off-target binding or neuroinflammatory effects cannot be ruled out. However, the co-localization of [^18^F]ACI-12589 signals in AD in vivo and in PSP with [^18^F]RO948 (a tau–PET ligand) may reflect off-target binding to neurodegenerative processes downstream of tau [103]. The exact properties of the binding of [^18^F]ACI-12589 in different NDDs will be the focus of future work. In summary, [^18^F]ACI-12589 has the ability to distinguish MSA cases from controls and other NDDs. However, the number of participants was relatively small, and further studies with larger patient cohorts are warranted in the future. Whether they are suitable for more common synucleinopathies, such as PD and DLB, remains to be investigated.

## 4. Exploration of Representative Compounds’ Scaffolds and Analogs

### 4.1. Phenothiazine Derivatives

The development of selective PET radiotracers targeting α-syn fibrils necessitates the rigorous exploration and structural optimization of novel lead compounds. Yu et al. identified a series of phenothiazine derivatives as promising candidates, enhancing their affinity and selectivity to α-syn fibrils through systematic scaffold modifications. In vitro binding assays revealed that (3-iodoallyl)oxy-phenothiazine (SIL23) exhibited greater binding affinity to recombinant α-syn fibrils than other SIL analogs [105]. Furthermore, [^125^I]SIL23 was shown to bind specifically to α-syn fibrils in postmortem brain tissues from PD patients and α-syn transgenic mice, with a high density of binding sites in brain tissues, suggesting its potential utility as a high-affinity ligand for in vivo imaging [105]. However, the researchers of SIL23 highlighted that its affinity for α-syn and selectivity over Aβ and tau fibrils remain insufficient for imaging α-syn fibrils in vivo [105]. Nevertheless, SIL23 still serves as a valuable tool for screening and identifying other ligands with improved affinity and selectivity. Guided by this approach, the team discovered other compounds, such as 3-methoxy-7-nitro-10H-phenothiazine (**2a**, K_i_ = 32.1 ± 1.3 nM) and 3-(2-fluoroethoxy)-7-nitro-10H-phenothiazine (**2b**, K_i_ = 49.0 ± 4.9 nM), which were subsequently radiosynthesized as [^11^C]**2a** and [^18^F]**2b**, respectively. Both tracers show specific binding to α-syn fibrils, high uptake across the BBB, and rapid clearance [106]. In vivo microPET imaging in healthy cynomolgus macaques further revealed that [^11^C]**2a** is uniformly distributed and rapidly washed out of the brain [106]. However, these compounds have not yet been evaluated in human studies, and further investigations are needed to assess their clinical potential.

### 4.2. Indolinone and Indolinone-Diene Analogs

Chu et al. conducted a structure‒activity relationship (SAR) study using the indolinone analog 3-(4-(dimethylamino)benzylidene)indolin-2-one (compound **5**) as a lead compound, synthesizing a series of novel 3-(benzylidene)indolin-2-one derivatives and evaluating their binding to α-syn, Aβ, and tau fibrils in vitro [107]. Most of the compounds exhibited only modest affinity for α-syn and lacked selectivity among the three fibrils. Researchers have systematically explored the impact of substituent modifications within the indolinone ring system. The introduction of a 4-nitrobenzene ring into the diene group facilitated the separation of stable Z, E and E, E regioisomers, with the Z, E configuration identified as the most pharmacologically active isomer. The substitution of the indolinone nitrogen with an N-benzyl group resulted in increased binding to α-syn and reasonable selectivity for α-syn over Aβ and tau [107]. The most potent and selective compound in the series was (Z)-1-(4-(2-fluoroethoxy)benzyl)-3-((E)-3-(4-nitrophenyl)allylidene)indolin-2-one (compound **46a**) (K_i_ = 2 nM), with 70-fold selectivity over Aβ and 40-fold selectivity over tau fibrils [107]. However, [^18^F]**46a** is considered unsuitable for in vivo imaging because of its high lipophilicity and potential for reducing the nitro group to the primary amino group. Consequently, compound **46a** serves as a secondary lead for further SAR exploration. Further optimization of the indolinone scaffold yielded [^18^F]WC58a with an affinity of 9 nM for α-syn. However, this compound was also characterized as too lipophilic (logP = 4.18) and too slowly cleared from the nonhuman primate brain to be evaluated in vivo [107].

### 4.3. Chalcone Analogs

Ono et al. identified a series of radioiodinated chalcone analogs, including flavonoids, chalcones, and aurones, that could inhibit the formation of Aβ [108,109,110] and α-syn fibrils [111,112], positioning these scaffolds as promising candidates for α-syn-targeted probe design. A subsequent SAR study revealed that modulating the conjugated double-bond length and the resulting molecular planarity critically influenced the binding affinity to α-syn aggregates [113]. The most potent compound was (2E,4E,6E,8E)-9-(4-(dimethylamino)phenyl)-1-(4-iodophenyl)nona-2,4,6,8-tetraen-1-one (IDP-4), which had the highest affinity and most selective binding to α-syn aggregates (K_d_ = 5.4 nM) compared with SIL-series tracers. The fluorescence staining of PD brain sections confirmed the affinity of IDP-4 for Lewy bodies [113]. However, ^125^I-IDP-4 has demonstrated suboptimal pharmacokinetics, with low brain uptake and slow clearance, making its use as a promising α-syn imaging probe difficult. Further SAR optimization of the chalcone scaffold produced a 50:50 mixture of isomers, (E)-3-(4-methoxystyryl)-5-(thiazol-2-yl)isoxazole and (E)-5-(4-methoxystyryl)-3-(thiazol-2-yl)isoxazole (compounds **11a**, **b**), with an α-syn affinity of 18.5 nM [114]. Recently, Ono’s group designed and synthesized a series of chalcone analogs with different aromatic modifications to evaluate their potentials as α-syn imaging probes [115]. They systematically evaluated aromatic modifications at the R2 position, identifying 4-(dimethylamino)phenyl substitutions as dual α-syn/Aβ binders. In fluorescence staining experiments, only the 4-nitrophenyl chalcone analog could successfully and selectively detect α-syn aggregates against Aβ aggregates in patient brain samples [115]. In addition, the radiotracer (2E,4E,6E)-7-(4-nitrophenyl)-1-(4-iodophenyl) hepta-2,4,6-trien-1-one ([^125^I]PHNP-3) exhibited high binding affinity for α-syn aggregates (K_d_ = 6.9 nM) and selectivity over Aβ [115], with moderate BBB permeability (0.78% ID/g at 2 min after intravenous injection). While the further optimization of pharmacokinetic properties is necessary, these results highlight the promising potential of the chalcone scaffold for α-syn-targeted imaging applications. Furthermore, the team designed and synthesized two new chalcone analogs, (2E,4E,6E)-7-(4-nitrophenyl)-1-[4-(2-hydroxy-3-fluoropropoxy)phenyl]hepta-2,4,6-trien-1-one (FHCL-1) and (2E,4E,6E)-7-(4-nitrophenyl)-1-[4-(2-hydroxy-3-fluoropropoxy)pyridinyl]hepta-2,4,6-trien-1-one (FHCL-2), and evaluated them via the central nervous system multiparameter optimization (CNS MPO) algorithm [116]. Both tracers exhibited high binding affinities for α-syn aggregates (K_i_ = 2.6 and 3.4 nM, respectively), with negligible Aβ binding [116]. Biodistribution studies in normal mice revealed enhanced brain uptake (2.09–2.40% ID/g at 2 min) compared with that of [^125^I]PHNPs [116]. In vivo autoradiography experiments confirmed the ability of [^18^F]FHCL-2 to detect α-syn aggregates in mouse models [116]. These preclinical studies validate the effectiveness of designing α-syn-targeted probes on the basis of CNS MPO scores and highlight the translational potential of ^18^F-labeled chalcone derivatives for α-syn PET imaging.

### 4.4. Quinoline and Bisquinoline Derivatives

Quinoline and bisquinoline derivatives have been extensively investigated as imaging probes for Aβ and tau aggregates [117,118,119]. Notably, a quinoline scaffold bearing a styrene group at the 2-position demonstrated high binding affinity to α-syn aggregates in vitro, with the conjugated double bond enhancing α-syn aggregation [120]. Ono’ team tried to develop α-syn probes by developing new scaffolds and adjusting the SAR. Their initial studies utilized small molecules to assess binding affinity via the suppression of ThT fluorescence upon interaction with α-syn aggregates. Among them, bis(8-hydroxy-2-quinolinylidene)azine (KPTJ10017), which contains a bisquinoline skeleton, reduced the fluorescence intensity to 62.1% of that of the controls and was considered a promising lead compound for targeting α-syn aggregates. On the basis of this scaffold, the team further designed and synthesized the bisquinoline derivatives (8-fluoroethoxy-2-quinolinylidene) (8-methoxy-2-quinolinylidene)azine (BQ1) and (8-fluoroethoxy-2-quinolinylidene) (8-hydroxy-2-quinolinylidene)azine (BQ2) [121], both of which showed high affinities for recombinant α-syn aggregates in vitro (K_i_ values of 17.0 and 11.6 nM, respectively) and moderate affinities for Aβ aggregates (K_i_ values of 8.5 and 7.3 nM, respectively). BQ2 demonstrated superior selectivity for α-syn aggregates over Aβ aggregates than BQ1 did in vivo. Both derivatives can clearly detect α-syn in human brain slices [121]. Biodistribution experiments revealed moderate uptake levels of [^18^F]BQ2 in the brain (1.59% ID/g 2 min after injection), supporting its potential for further optimization as an α-syn PET tracer [121]. Recently, they evaluated 16 quinoline and quinoxaline derivatives for α-syn imaging suitability. The quinoline skeleton is superior to the quinoxaline skeleton in the binding of α-syn aggregates. In addition, the position and type of substituents significantly affect the binding affinity to α-syn and Aβ aggregates [122]. Among them, the quinoline derivative SQ3, featuring p-(dimethylamino)phenylvinyl and fluoroethoxy groups at positions 2 and 7, displayed moderate selectivity for α-syn aggregates (K_i_ = 230 nM) over Aβ aggregates and maintained a high binding affinity for α-syn aggregates (K_i_ = 39.3 nM) [122]. In a biodistribution experiment, [^18^F]SQ3 showed high uptake in the brains of normal mice (2.08% ID/g 2 min after intravenous injection), indicating the strategic utility of quinoline-based architectures in α-syn-targeted tracer design [122].

### 4.5. Benzimidazole (BI) Derivatives

Watanabe designed and synthesized three novel radioiodinated BI derivatives for imaging α-syn aggregates in the brain. Among the derivatives tested in their study, [^125^I]2-((1E,3E)-4-(iodophenyl)buta-1,3-dien-1-yl)-1-(4-methoxybenzyl)-1H-benzo[d]imidazole ([^125^I]BI-2) exhibited the highest selective binding affinity for α-syn aggregates [123]. BI-2 exhibited superior or comparable α-syn/Aβ selectivity to IDP and phenothiazine derivatives (e.g., SIL5, SIL23, and SIL26) but lower selectivity than certain 3-(benzyl)indolin-2-one analogs, including 46a [123]. Furthermore, BI-2 has a low initial uptake (0.56% ID/g) and poor elution kinetics in the normal mouse brain, restricting its utility for in vivo imaging [123]. In summary, further structural modifications of BI derivatives may yield viable α-syn-targeted tracers.

### 4.6. Phenylbenzofuranone (PBF) Derivatives

Akasaka designed novel radioiodinated PBF derivatives ((Z)-2-{(E)-3-[4-(Dimethylamino)phenyl]allylidene}-5-iodobenzofuran-3(2H)-one (IDPBF-2), (Z)-5-Iodo-2-[(E)-3-(4-nitrophenyl)allylidene]-benzofuran-3(2H)-one (INPBF-2), (Z)-2-{(2E,4E)-5-[4-(Dimethylamino)phenyl]penta-2,4-diene-1-ylidene}-5-iodobenzofuran-3(2H)-one (IDPBF-3), and (Z)-5-Iodo-2-[(2E,4E)-5-(4-nitrophenyl)penta-2,4-diene-1-ylidene]-benzofuran-3(2H)-one (INPBF-3)) on the basis of dimethylamino/nitro-substituted scaffolds and evaluated their potential as α-syn imaging probes [124,125]. In vitro competition assays revealed that the K_i_ values of the four compounds against α-syn aggregates were 4.1, 3.4, 0.37, and 0.28 nM, respectively [126]. The extended conjugated double bonds in PBF derivatives enhanced α-syn selectivity over Aβ, with the nitro-substituted derivative (INPBF-3) demonstrating superior α-syn/Aβ selectivity compared with the dimethylamino-substituted derivative (IDPBF-3) [126]. These results suggest that the PBF scaffold may be useful for the development of α-syn imaging probes with high binding affinity to α-syn aggregates. In addition, it may be important to elongate the conjugated double bonds and insert nitro groups into the molecules to develop better α-syn imaging probes. However, all the radioiodinated PBF derivatives exhibited insufficient brain uptake due to their high lipophilicity. While PBF derivatives exhibit foundational characteristics for α-syn imaging, further structural modifications are needed to achieve applicability in vivo.

### 4.7. Indole Derivatives

Zeng et al. designed and synthesized a series of cyano-substituted indole derivatives through SAR studies. Their preliminary screening results revealed that 2-((1-benzyl-1H-indol-3-yl)methylene)malononitrile (compound 23) exhibited moderate binding affinity to α-syn fibrils (K_i_ = 3.5 ± 0.8 nM) and moderate selectivity for Aβ_1–42_ fibrils (4.2-fold) [127]. They introduced an iodine atom and converted compound 23 into the radioiodinated ligand 2-((1-(4-iodobenzyl)-1H-indol-3-yl)methylene)malononitrile ([^125^I]51), which demonstrated improved α-syn selectivity (K_i-α-syn_ = 17.4 nM, K_i-Aβ_ = 73.0 nM), moderate uptake (brain uptake_2 min_ = 3.57 ± 0.28% ID/g), and good clearance (brain uptake_2 min_/brain uptake_60 min_ = 7.28) in normal mice [127]. While this scaffold provides a novel structural framework for α-syn tracer development, further chemical refinement is needed to enhance target engagement and pharmacokinetic profiles.

### 4.8. Pyridothiophene Derivatives

Recent works by Pees et al. have synthesized 47 pyridothiophene derivatives, identifying (*R*)-5-(6-(3-fluoropyrrolidin-1-yl)pyridin-3-yl)-2-(pyridin-3-yl)thieno[3,2-b]pyridine (asyn-44) as a lead candidate that demonstrates good imaging properties in vitro because of its high affinity for PD tissue homogenates (K_d_ = 1.85 ± 0.38 nM) and low affinity for AD tissues (K_d_ = 170 ± 60 nM) [128]. ARG using [^3^H]asyn-44 revealed co-localization with anti-pS129 immuno-reactivity in MSA and PD brain sections, confirming its specificity for pathological α-syn, with minimal binding observed in AD, PSP, corticobasal degeneration (CBD) pathology, and control cases [128]. Radiolabeling with fluorine-18 yielded [^18^F]asyn-44, which exhibited favorable brain permeability and moderate clearance in wild-type rats [128]. This work suggests that asyn-44 is an effective compound and that the pyridothiophene scaffold is promising for the development of α-syn PET tracers for PD and MSA. Owing to the presence of radioactive metabolites in the rat brain, this study was not further performed for PET imaging in rodents.

### 4.9. N,N-Dibenzylcinnamamide (DBC) Derivatives

Chen and colleagues designed a series of N,N-DBC derivatives as novel PET imaging probes for α-syn aggregates via a high-throughput surface plasmon resonance (SPR)-based screening platform [125]**.** Structural and SAR analyses of these compounds revealed a high-affinity tracer (5–41) with high binding affinity (K_d_ = 1.03 nM) [125]. The SPR platform enables the simultaneous evaluation of binding potency, selectivity across multiple protein targets, and high-throughput SPR technology. The fluorine-modified DBC derivatives retained excellent affinity, suggesting the possibility of developing F-labeled probes for α-syn PET imaging [125]**.**

### 4.10. Arylpyrazolethiazole (APT) Derivatives

Bonanno reported a series of APT derivatives in which the benzene ring of dipeptidyl peptidase analogs was replaced with a thiazole moiety, significantly reducing lipophilicity [129]. After modification with different substituents, 4-(5-(4-bromothiazol-2-yl)-1H-pyrazol-3-yl)-N-methylaniline (APT-13) exhibited α-syn fibril affinity (K_i_ = 27.8 ± 9.7 nM) and >3-fold selectivity over Aβ (K_i_ = 92.6 ± 48.8 nM) in vitro [129]. Radiolabeled [^11^C]APT-13 showed excellent brain permeability and brain uptake in mice (SUV_peak_ = 1.94 ± 0.29) and rapid clearance from the brain (t_1/2_ = 9 ± 1 min) [129]. Therefore, the APT scaffold is a good candidate for low-lipophilicity α-syn PET tracers, although future studies require validation in human pathological brain tissue and imaging in fibril-injected rodent models.

### 4.11. 2,6-Disubstituted Imidazo[2,1-b][1,3,4]thiadiazole (ITA) Derivatives

Zeng et al. developed 2,6-disubstituted ITA as a new α-syn-binding scaffold [130]. Through ARG studies, researchers discovered a lead iodide compound, 6-(4-iodophenyl)-2-(pyrazin-2-yl)imidazo[2,1-b][1,3,4]thiadiazole (ITA-3, compound **48**), with a pyrazin-2-yl group which had moderate binding affinity (IC_50_ = 55 nM) and satisfactory BBB permeability (brain_2 min_ = 4.9% ID/g) to α-syn pathology in human PD brain slices [130]. Structural optimization yielded potential fluorinated 6-(6-fluoropyridin-3-yl)-2-(pyrazin-2-yl)imidazo[2,1-b][1,3,4]thiadiazole (compound **68**) based on [^125^I]ITA-3, namely, [^18^F]FITA-2, which showed high initial brain uptake (SUV_peak_ = 2.80 ± 0.45), good clearance, and good stability for α-syn detection in postmortem human PD brain tissue sections [130]. [^18^F]FITA-2 is currently being evaluated in patients, and further structural optimization is being carried out to improve its affinity and selectivity.

### 4.12. 2-Pyridone Analogs

FN075 is a dihydrothiazolo ring-fused 2-pyridone peptidomimetic with a free carboxylic acid that was initially studied for its antibiotic activity via the inhibition of bacterial amyloid fibril formation [131]. Paradoxically, FN075 accelerated the onset of α-syn fibrillization in vitro [132]. A study in which α-syn-expressing flies were fed FN075 revealed that the α-syn-FN075 interaction stimulated α-syn formation in vitro [133]. Intranigral FN075 injection in mice induced tyrosine hydroxylase (TH)-positive neuronal loss and PD-like motor deficits at 3 months post-injection [134]. On the basis of these findings, the SAR of FN075 was investigated, indicating that the carboxylic acid group of FN075 may be a key part of its activity. Aberg et al. developed FN075-derived PET tracers, and imaging in healthy nonhuman primates (NHPs) revealed that [^11^C]**84** contains a carboxylic acid group and displays negligible brain uptake in nonhuman primates. The derivative [^11^C]**85**, which was produced by masking the carboxylic acid group as an acetoxymethyl ester, achieved brain entry and exhibited slow washout, suggesting that acetoxymethyl ester is unstable in vivo [135].

Cairns et al. later reported the first NHP PET imaging of radiolabeled FN075 analogs, employing a prodrug strategy to increase brain delivery [136]. The authors synthesized and ^11^C-radiolabeled 2-pyridone (3R)-7-([4′-methoxynapthylen-1-yl]methyl)-5-oxo-8-(3-(trifluoromethyl)-phenyl)-2,3-dihydro-5H-thiazolo[3,2-a]pyridine- 3-carboxylic acid (compound **12**) ([^11^C]**12**) and its acetoxymethyl ester analog (3R)-7-([4′-methoxynapthylen-1-yl]methyl)-5-oxo-8-(3-(trifluoromethyl)-phenyl)-2,3-dihydro-5H-thiazolo[3,2-a]pyridine- 3-carboxylic acid acetoxymethyl ester (compound **14**) ([^11^C]**14**). The PET imaging of NHPs revealed that [^11^C]**14** achieved four-fold greater brain uptake (SUV_peak_ = 0.8) than [^11^C]**12** did, validating the prodrug approach to circumventing the inherent BBB impermeability of 2-pyridones [136]. As expected, at the beginning of this study, 2-pyridones did not cross the BBB efficiently. While the lipophilic profile of FN075 limits its utility as an imaging biomarker, the prodrug derivatization strategy is broadly applicable to related scaffolds. Future development will focus on improving the physicochemical properties and optimizing their affinity and selectivity for aggregated α-syn relative to other protein aggregates [136].

### 4.13. Benzothiazole-Ethenyl-Phenol Derivative (F0502B)

F0502B represents a promising lead compound that was recently obtained by Ye’s team through screening, organic synthesis, and the further optimization of the top structural backbones, followed by counter-screening using in vitro fibrils and primary neurons with α-syn, Aβ, and tau aggregates [137,138]. This study demonstrated a PET tracer design strategy by targeting potential conserved binding sites shared by in vitro and ex vivo (human) fibrils. The results revealed that F0502B specifically recognized α-syn fibrils in vitro and in vivo, with SAR studies revealing that the phenol group was necessary for F0502B to selectively bind to α-syn aggregates and that its destruction decreased its binding affinity to α-syn fibrils [137]. Radiolabeling with fluorine-18 yielded [^18^F]-F0502B, which exhibited a much lower affinity for Aβ_1-42_ (K_d_ = 109.2 nM) or tau fibrils (K_d_ = 120.5 nM) than for α-syn fibrils (K_d_ = 10.97 nM) in in vitro binding assays. [^18^F]-F0502B also shows a high affinity for brain homogenates from PD and DLB patients but has no specific binding affinity for AD or healthy patients [137]. In vivo, α-syn aggregates in the brains of mice, macaques, and PD patients can also be recognized. In addition, F0502B has good brain permeability and is rapidly removed from normal brains [137]. However, while ARG studies have shown that F0502B interacts with α-syn aggregates in postmortem brain slices, it remains unclear whether it enables the imaging of α-syn pathology in living patients, necessitating further translational validation [139]. Table 2 summarizes the characterization of representative compounds’ scaffolds and analogs, and their chemical structures are listed in Figure 2.

## 5. Identifying α-Syn Radioligands Through High-Throughput Screening (HTS)

### 5.1. [^125^I]21

HTS and computational modeling have emerged as strategic tools for identifying α-syn-targeted ligands. Mach’s group previously developed a new paradigm for the development of radioligands for α-syn fibrils via HTS, identifying the lead 2-(3,4-dimethylphenoxy)-N-(3-(4-iodophenyl)isoxazol-5-yl)acetamide (compound **61**) [140]. Radiolabeled [^125^I]**61** exhibited nanomolar affinity for α-syn fibrils (K_d_ = 1.06 nM) and moderate Aβ_1–42_ fibril binding (K_d_ = 4.56 nM) [140]. ARG studies detected [^125^I]**61** (1 nM) binding in sarkosyl-insoluble fractions of 15-month-old A53T transgenic mouse brains, although its suboptimal physicochemical properties limit its utility in whole-brain-section analyses [140]. Subsequent HTS efforts focused on isoxazole derivatives, yielding N-(3-(4-iodophenyl)isoxazol-5-yl)-4-methoxybenzamide (compound **21**) with enhanced α-syn affinity (K_d_ = 0.48 ± 0.08 nM) and Aβ_42_ fibrils (K_d_ = 2.47 ± 1.30 nM) and improved selectivity over Aβ_42_ [141]. The preclinical PET imaging of α-syn preformed fibril (PFF)-injected mice revealed the increased retention of [^11^C]**21** compared with that in control mice. In addition, healthy NHPs show high nonspecific binding, although they also show high initial brain uptake and rapid clearance [141]. However, the relatively high binding of [^11^C]**21** to Aβ fibrils and AD brain tissues limits its clinical applicability, as it is present in approximately 50% of PD patients at autopsy [142]. While this work validates SAR studies and in silico HTS for α-syn tracer discovery, clinically translatable probes require enhanced selectivity for α-syn over Aβ/tau aggregates.

### 5.2. 2FBox and 4FBox

Verdurand et al. employed bioinformatics modeling to prioritize α-syn-targeted candidates, synthesizing and evaluating two candidates, N-{4-[(E)-2-(1,3-Benzoxazol-2-yl)ethenyl]phenyl}-2-fluoro-N-methylbenzene-1-sulfonamide (2FBox) and N-{4-[(E)-2-(1,3-Benzoxazol-2-yl)ethenyl]phenyl}-4-fluoro-N-methylbenzene-1-sulfonamide (4FBox), with the highest docking scores for the α-syn binding pocket [143]. In vitro assays revealed that 2FBox exhibited an approximately 47-fold greater α-syn affinity and 44-fold greater α-syn/Aβ selectivity than did 4FBox. However, both failed as ligands for α-syn PET radiotracers [143]. Through ARG experiments, [^18^F]2FBox and [^18^F]4FBox failed to detect α-syn in TgM83 mice or in LBs or GCIs from postmortem PD/MSA brain sections [143]. PET imaging in a rat model also revealed that both candidate radioligands failed to detect Aβ or α-syn fibrils in vivo despite crossing the BBB [143]. This study highlights the limitations in computational prediction accuracy for α-syn tracer development, necessitating refined computational modeling strategies to bridge in silico design and in vivo efficacy.

### 5.3. 4,4′-Disarylbisthiazole (DABTA) Scaffold

Recent efforts have focused on 4,4′-DABTA-scaffold-based radiotracers, where the structural incorporation of a methylenedioxy group enhances α-syn binding affinity and selectivity [144]. After the binding sites and molecular properties of α-syn were predicted through computer modeling and machine learning, Uzuegbunam’s team constructed a focused DABTA library, with four candidate tracers exhibiting subnanomolar α-syn affinity (K_d_ = 0.10–1.22 nM) and >200-fold selectivity over Aβ/tau aggregates. Among them, 4-(benzo[d][1,3]dioxol-5-yl)-4′-(6-[18F] fluoropyridin-3-yl)-2,2′-bithiazole ([^18^F]d6) and 6-(4′-(6-[18F]fluoropyridin-3-yl)-[2,2′-bithiazol]-4-yl)-[1,3]dioxolo[4,5-b]pyridine ([^18^F]d8) exhibited optimal brain pharmacokinetic properties, with [^18^F]d8 showing excellent brain uptake and metabolic stability (no brain metabolites detected) in mouse dynamic PET imaging experiments [145]. Owing to these excellent binding and metabolic properties, [^18^F]d8 is considered a clinically promising candidate, pending validation in postmortem human NDD brain tissues via ARG studies.

### 5.4. Anle138b Analogs

Wagner’s previous research developed a novel oligomer modulator, anle138b [3-(1,3-benzodioxol-5-yl)-5-(3-bromophenyl)-1H-pyrazole], on the basis of systematic HTS combined with medicinal chemistry optimization for disease-modifying treatments of NDDs [146]. In PD and MSA mouse models [147,148], anle138b binds α-syn aggregates with moderate affinity (K_d_ = 190 ± 120 nM) [149] and exhibits favorable BBB penetration and oral bioavailability [150,151]. Despite its therapeutic promise, anle138b’s high lipophilicity (calculated logP = 4.34) limits its utility as a PET tracer.

The anle138b analog (3-(3-bromophenyl)-5-[4-(dimethylamino)phenyl]-1H-pyrazole) (**anle253b)** was developed as a PET tracer that could target α-syn fibrils, demonstrating high in vitro affinity (IC_50_ = 1.6 nM) and preferential binding to fibrils over oligomers/monomers [152]. Despite crossing the BBB, [^11^C]anle253b exhibited low brain uptake and unfavorable pharmacokinetics in rats, which were attributed to its high lipophilicity (clogP = 5.21) [152]. To address this, the pyridine-modified derivative 4-[3-(4-dimethylaminophenyl)-1H-pyrazol-5-yl]-2-bromopyridine (MODAG-001) was synthesized by modifying the chemical structure of anle253b, where the phenyl group was exchanged with pyridine to reduce the lipophilicity to a clogP of 3.85 and improve the binding affinity [124]. In in vitro binding assays, [^3^H]MODAG-001 showed high binding affinity to pure recombinant α-syn (K_d_ = 0.6 ± 0.1 nM), with 30-fold selectivity over Aβ/tau fibrils [124]. The preclinical evaluation of [^11^C]MODAG-001 in mice revealed good BBB permeability, improved brain uptake (SUV_peak_ = 1.4), and relatively rapid clearance from the brain, although ARG failed to detect α-syn aggregates in human brain sections from DLB patients [124]. [^11^C]MODAG-001 exhibited good brain uptake in murine PET imaging, although radioactive metabolites from its demethylated form hindered quantification owing to BBB penetration. To inhibit the metabolic demethylation process, a deuterated analog of [^11^C]MODAG-001, (d3)-[^11^C]MODAG-001, was designed. PET imaging revealed striatal binding in α-syn-PFF-injected rats, with a peak SUV of 2.1. However, two radioactive metabolites from radiotracers were detected in the plasma and brain after the dynamic whole-body PET imaging of the mice [124]. Further evaluation of (d3)-[^11^C]MODAG-001 including the intrastriatal injection of α-syn-PFFs or postmortem human AD/DLB brain homogenates from pigs revealed binding to α-syn fibrils, but no significant uptake was observed in DLB homogenates, possibly due to low α-syn concentrations or morphological differences in pathological aggregates and nonspecific binding in AD homogenates [153,154]. In addition, the intracerebral injection model failed to simulate intracellular inclusions due to the short injection time, and the injected α-syn-PFF concentration was much higher than the actual concentration in the diseased human brain and was only used as a proof of concept [155]. Nevertheless, MODAG-001 is still a suitable lead molecule for further radioligand development and evaluation because of its suitable pharmacokinetic and biodistribution properties, which are contingent on enhancing signal-to-noise ratios and pathological specificity [124].

### 5.5. N-Phenylbenzamide Analogs

Borroni and colleagues described radiolabeled N-phenylbenzamide analogs in their patent research, with [^3^H]BF2846 exhibiting high binding affinity (K_d_ = 2.0 nM) to site 9 of α-syn fibrils in radioligand binding assays and ARG studies [75,156]. These analogs reportedly have good target engagement, favorable brain penetration, and low nonspecific binding in the brain, but ^18^F-labeled BF2846 analogs are limited by Aβ fibril cross-reactivity. HY-2-15 was identified from SAR studies based on BF2846 and was radiolabeled with tritium and ^11^C, which resulted in a high α-syn affinity (K_d_ = 6.1 ± 2.1 nM) and low binding affinity to Aβ fibrils (K_d_ = 33–115 nM) [157]. In vitro autoradiography studies revealed that [^3^H]HY-2-15 specifically binds to α-syn in MSA (K_d_ = 4.5 nM) and PD (K_d_ = 5.1 nM) postmortem tissues, although residual binding to tauopathies (CBD: 18 nM; PSP: 7.1 nM) suggested potential tau aggregate interactions. Nonhuman primate PET studies confirmed that [^11^C]HY-2-15 achieved moderate brain uptake (SUV ~1.6) and rapid washout [157]. These properties position [^11^C]HY-2-15 as a candidate tracer for α-syn imaging in MSA, pending the validation of its pathological specificity in clinical studies.

Kim and colleagues performed an SAR study using BF2846 as a lead compound, replacing the piperazine moiety of BF2846 with a 3.8-diazabicyclo[3.2.1]octane moiety, identifying the promising candidate **4i** (4-methoxy-N-(4-(3-(pyridin-2-yl)-3,8-diazabicyclo[3.2.1]octan-8-yl)phenyl)benzamide) [158]. Synthetic CLX**4i** showed crosslinks primarily adjacent to site 9 (DMPVDPDNE, residues 115–123), with secondary crosslinks identified at site 2 (GVVHGVATVAEKTKE, residues 47–61) and the N-terminus (EGVVAAEK, residues 15–22). In vitro assays revealed that **4i** exhibited nanomolar binding affinity (K_i_ = 6.1 nM) for recombinant α-syn fibrils and low affinity for Aβ fibrils in AD homogenates. **4i** also exhibited highly specific binding to AD, progressive supranuclear palsy, and corticobasal degeneration tissues, as well as PD and MSA tissues, indicating significant affinity for tau [158]. Nuclear emulsion autoradiography of [^3^H]**4i** in tissue sections demonstrated that [^3^H]**4i** may have a strong binding affinity to GCIs rather than LBs in MSA. PET studies on NHPs confirmed that [^11^C]**4i** shows good brain penetration and rapid washout. In research on metabolism, [^11^C]**4i** was rapidly metabolized to polar metabolites that did not bind to α-syn fibrils [158]. Overall, [^11^C]**4i** may have potential as a PET radiotracer for imaging α-syn in patients with MSA. However, its high binding affinity for tau may limit the utility of [^11^C]**4i** in some cases. However, the absence of tau in the cerebellum suggests that future imaging of GCIs in control subjects and subjects with the cerebellar form of MSA may be possible. Table 3 provides an overview of α-syn radioligands identified through high-throughput screening, and the chemical structures of the α-syn radioligands are listed in Figure 3.

## 6. Conclusions and Future Perspectives

The development of PET radiotracers for α-syn aggregates with appropriate pharmacokinetics, high binding affinities, and specific selectivity has achieved marked progress over the past decade, although major challenges remain. These advances are highly valuable for the early diagnosis, disease course monitoring, and evaluation of therapeutic efficacy in synucleinopathies [75]. Several radiolabeled probes have been explored for the imaging of α-syn pathology in the brain, including scaffolds and derivatives derived from AD, as well as other representative compound scaffolds and analogs and the validation of ligands derived from HTS. Most of the compounds discussed here demonstrated nanomolar affinity for α-syn aggregates alongside BBB permeability and uniform brain distribution in preclinical models in vivo. However, no α-syn PET tracer has been approved for clinical use, with current candidates facing persistent challenges in distinguishing α-syn from other pathological proteins such as Aβ and Tau. Considering the high prevalence of mixed proteinopathies in neurodegenerative disorders, this is a critical limitation. Emerging candidates, such as (d3)- [^11^C]MODAG-001 and [^18^F]F0502B demonstrated high affinity and favorable brain kinetics in experimental animal models. ^18^F-SPAL-T-06 and [^18^F]ACI-12589 showed high affinity and selectivity in imaging α-syn relative to Aβ in MSA patients. ^18^F-C05-05 exhibited encouraging results in both PD and MSA-P patients. Although they are not ideal for widespread use in imaging synucleinopathies, some of them have achieved encouraging results in human imaging research. This review aimed to promote the progress of neuropathology by critically evaluating the current state of development of PET tracers for α-syn, moving toward preventive measures and early diagnosis in the field of α-syn with greater accuracy and accessibility.

Future efforts could prioritize a multimodal neuroimaging approach, integrating small-molecule ligands for total α-syn aggregate detection and antibody-based probes for extracellular pathology monitoring. Small-molecule ligands have advantages in terms of BBB penetration and rapid kinetics, whereas antibodies have subnanomolar specificity despite limited brain bioavailability (~0.1% uptake) and slow pharmacokinetics [159,160]. HTS may help to identify α-syn ligands, and in silico modeling design during routine tracer development could accelerate the identification of new lead compounds for specific α-syn PET imaging [156]. Additionally, structure-based drug design seems ideal but is hampered in practice because the high-resolution crystal structure of α-syn fibrils remains elusive. In future studies, small-molecule ligands may be more suitable for imaging total α-syn aggregates, whereas antibodies may be helpful for imaging extracellular aggregates and monitoring treatment response, which may complement each other [161]. Multimodal biomarker integration (e.g., combining PET with MRI or CSF analysis) may increase diagnostic accuracy for heterogeneous synucleinopathies such as PD and atypical PS [162]. These strategies, combined with the SAR optimization of structurally diverse small molecules (e.g., pyridothiophene, DABTA analogs), could bridge the gap between preclinical studies and clinical translation, enabling early diagnosis and targeted therapeutic evaluation.

## Figures and Tables

**Figure 1 cells-14-00907-f001:**
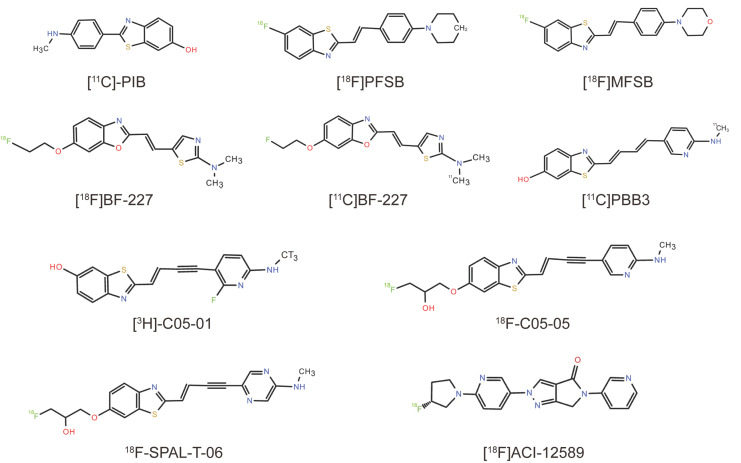
Chemical structures of radiolabeled compounds from AD.

**Figure 2 cells-14-00907-f002:**
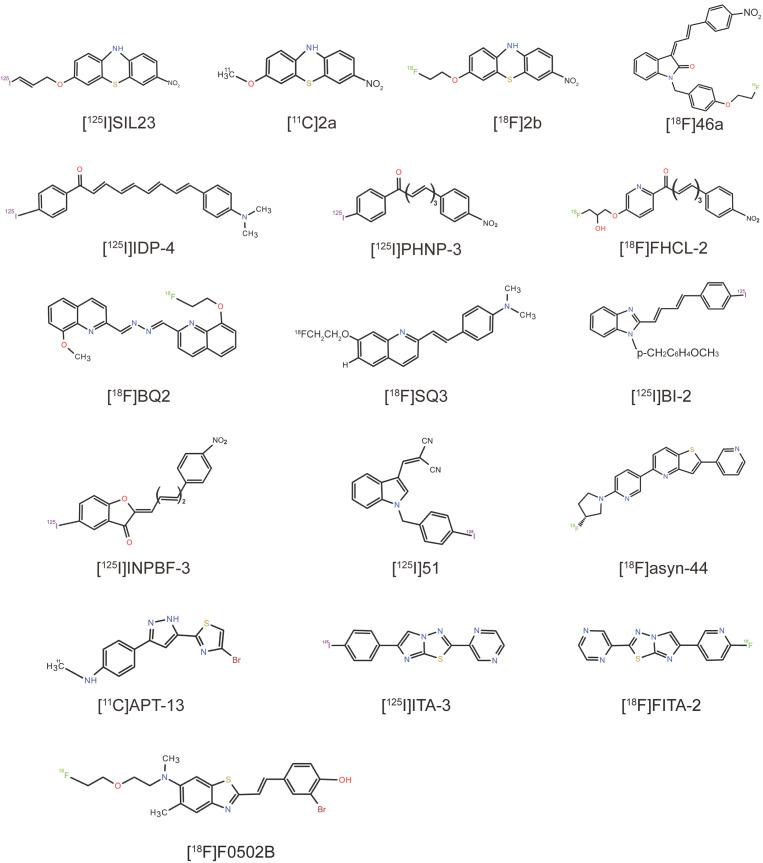
Chemical structures of representative compounds’ scaffolds and analogs.

**Figure 3 cells-14-00907-f003:**
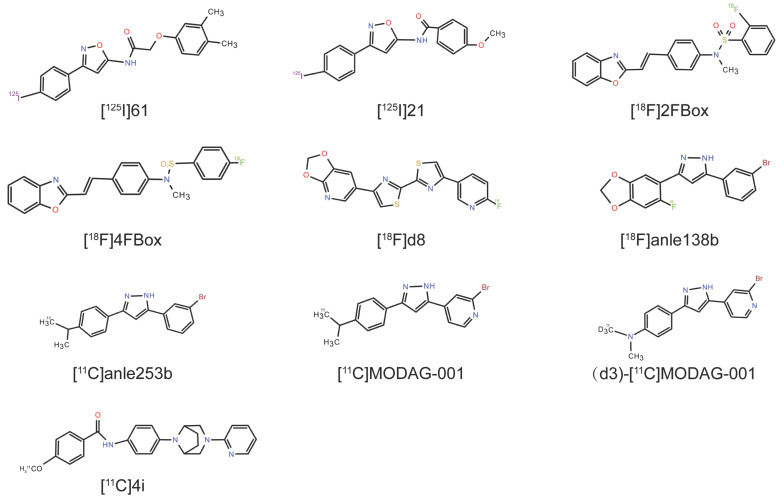
Chemical structures of α-syn radioligands.

**Table 1 cells-14-00907-t001:** Summary table of radiolabeled compounds from AD.

Radiolabeled Compounds	In Vitro Assays				Preclinical Trial					Clinical Trial			Advantages	Limitations
	K_i_ or k_d_ (α-Syn Fibrils) (nm)	K_i_ or k_d_ (aβ Fibrils) (nm)	K_i_ or k_d_ (tau Fibrils) (nm)	High Selectivity (α-Syn/aβ)	Rcy [%]	Am [gbq/μmol]	Bbb Permeability with Early Peak Uptake of Suv	≥0.4% id/g (Rat Brain) or ≥4.0% id/g (Mouse Brain) [%]	Clearance Rate	Pet Imaging in Animals	Pet Imaging in Human Subjects	Tested In Vitro Autoradiography (Gold Standard)		
[^11^C]-PIB	10.07	0.71	N.T.	No	N.T.	2.679	N.T.	N.T.	N.T.	N.T.	N.T.	N.T.	N.T.	Fails to bind to pure DLB (Aβ-free) brain homogenates
[^18^F]PFSB	25.4 ± 2.3	N.T.	N.T.	Yes	5.8 ± 1.3	36.5 ± 8.5	4.78	N.T.	N.T.	N.T.	N.T.	High binding in MSA and PD brain tissues; no binding in Aβ plaques and tau pathology	Selective binding to α-syn pathology on human brain slices	Highly lipophilic
[^18^F]MFSB	10.3 ± 4.7	N.T.	N.T.	Yes	11.6 ± 2.9	41.2 ± 12.0	Passing BBB, SUV = 1.79 ± 0.02	N.T.	Fast to moderate clearance	N.T.	N.T.	High specific binding in MSA brain slices and no increased binding in AD sample	High affinity and selectivity to α-syn over Aβ, less lipophilic	Pharmacokinetic profile requires some improvement to determine and reduce nonspecific binding
[^18^F]BF-227	9.63	1.31–80	N.T.	No	N.T.	40–840	Passing BBB, SUV_peak_ = 2.7	0.65	N.T.	N.T.	N.T.	Autoradiography does not support significant binding to GCI	High affinity for α-syn	No binding towards α-syn aggregates
[^11^C]BF-227	N.T.	N.T.	N.T.	No	>50, based on [^11^C]-CH3OTf)	119–138	Passing BBB, SUV_peak_ > 1.5	N.T.	More gradual clearance than in HC brains	N.T.	HCs (8); probable MSA (8)	N.T.	Binding to GCI-rich brain regions by PET study on patients	Contradictory results among different research groups concerning binding towards α-syn aggregates
[^11^C]PBB3	N.T.	N.T.	N.T.	No	N.T.	133	N.T.	N.T.	N.T.	N.T.	HCs (3); PSP-P (4); PSP-RS (1); DCTN1 mutation with PSP-P phenotype (1); MSA-P (1)	No significant binding in LB disease cases and significant binding to MSA cases	Usefulness of differentiating tauopathies from α-synucleinopathies	Low affinity for α-syn; cannot detect α-syn in LBD
[^3^H]-C05-01	25	N.T.	N.T.	No	N.T.	0.81	N.T.	N.T.	N.T.	N.T.	N.T.	Specific binding in cases with α-syn pathology	High affinity for α-syn from DLB patient	Relatively high nonspecific and off-target binding
^18^F-C05-05	N.T.	N.T.	N.T.	Yes	N.T.	63–557	Passing BBB, SUV = 1.11 − 2.5	N.T.	Slightly slower than PM-PBB3 and control	α-syn marmoset (1)	HCs (8); PD (8); DLB (2); MSA-P (3)	Intense binding to GCIs and DLB and PDD cases	High affinity for α-syn aggregates (IC_50_ = 8.0 nM); not highly binding with Aβ and tau aggregates in AD tissues (IC_50_ of 12.9 nM)	Not markedly penetrant through BBB; not sensitive to early-stage Lewy pathologies
^18^F-SPAL-T-06	2.49	N.T.	N.T.	Yes	N.T.	237.5 ± 53.9	N.T.	N.T.	Rapid clearance	N.T.	MSA-P (2); MSA-C (1); HC (1)	High binding with GCIs	High reactivity with MSA-type α-syn; negligible cross-reactivity with off-target components	Fails to capture PD and DLB pathologies
[^18^F]ACI-12589	33.5 ± 17.4	No binding	No binding	Yes	25.3 ± 4.5	11.1	Passing BBB with rapid brain uptake	N.T.	Rapid washout	N.T.	α-syn related disorders (23); other neurodegenerative disorders (11) and HCs (8)	Specific binding in cases with MSA, PD, PDD, and LBV-AD	Specific to α-syn and good selectivity	Fails to capture PD and DLB pathologies

RCY: radiochemical yield; AM: molar activity; PET: positron emission tomography; α-syn: alpha-synuclein; Aβ: β-amyloid; BBB: blood–brain barrier; SUV: standardized uptake value; GCIs: glial cytoplasmic inclusions; HCs: healthy controls; PD: Parkinson’s disease; PDD: Parkinson’s disease with dementia; DLB: dementia with Lewy bodies; LBD: Lewy body disorder; AD: Alzheimer disease; LBV-AD: Lewy body variant Alzheimer’s disease; PSP: progressive supranuclear palsy; PSP-P: PSP—Parkinsonism phenotype; PSP-RS: PSP—Richardson phenotype; MSA: multiple system atrophy; MSA-C: MSA phenotype dominated by cerebellar ataxia; MSA-P: MSA phenotype dominated by Parkinsonism; N.T.: not tested.

**Table 2 cells-14-00907-t002:** Summary table of representative compounds’ scaffolds and analogs.

Radiolabeled Compounds	In Vitro Assays				Preclinical Trial					Clinical Trial			Advantages	Limitations
	K_i_ or K_d_ (α-Syn Fibrils) (nM)	K_i_ or K_d_ (Aβ Fibrils) (nM)	K_i_ or K_d_ (tau Fibrils) (nM)	High Selectivity (α-Syn/Aβ)	RCY [%]	AM [GBq/μmol]	BBB Permeability with Early Peak Uptake of SUV	≥0.4% ID/g (Rat Brain) or ≥4.0% ID/g (Mouse Brain) [%]	Clearance Rate	PET Imaging in Animals	PET Imaging in Human Subjects	Tested In Vitro Autoradiography (Gold Standard)		
[^125^I]SIL23	120–180	635	230	Yes	43	81.4	N.T.	N.T.	N.T.	N.T.	N.T.	N.T.	High affinity and relative selectivity for α-syn, specific binding to insoluble protein from human PD brain samples	High lipophilicity (calculated logP = 5.7)
[^11^C]2a	32.1 ± 1.3	K_i-α-syn_/K_i-Aβ_ > 3 times	K_i-α-syn_/K_i-tau_ > 4 times	Yes	35–45	>363	Passing BBB with high initial uptake	0.953 ± 0.115	Rapid washout kinetics, faster washout kinetics than [^18^F]2b	Cynomolgus macaque (1)	N.T.	N.T.	High initial uptake and rapid washout	Moderate affinity
[^18^F]2b	49.0 ± 4.9	Ki_-α-syn_/K_i-Aβ_ = 2.1 times	K_i-α-syn_/K_i-tau_ = 2.5 times	Yes	55–65	>200	Passing BBB with high initial uptake	0.758 ± 0.013	Rapid washout kinetics	N.T.	N.T.	N.T.	High initial uptake and rapid washout	Moderate affinity
[^18^F]46a	8.9	271	50	Yes	N.T.	29.6–185	N.T.	N.T.	N.T.	N.T.	N.T.	N.T.	-	High logP value (4.18)
[^125^I]IDP-4	5.4 ± 1.5	12.9–37.1	N.T.	No	19–60	81.4	N.T.	0.45%	Slow washout	N.T.	N.T.	N.T.	Selective binding affinity for α-syn aggregates	Low brain uptake
[^125^I]PHNP-3	6.9	102 ± 21	N.T.	Yes	25	N.T.	Poor BBB penetration	0.78	N.T.	N.T.	N.T.	N.T.	Modest uptake; high affinity and selectivity	Low brain uptake; high lipophilicity; high molecular weight (431 Da)
[^18^F]FHCL-2	3.4	N.T.	N.T.	Yes	36	5.3	Passing BBB with high brain uptake	2.4	Gradual clearance	N.T.	N.T.	N.T.	High binding affinity for α-syn aggregates	Low 2/60 min ratios of radioactivity in brain (1.1–1.5)
[^18^F]BQ2	11.6	7.3	No	No	1.2	8.9	Passing BBB, moderate brain uptake	1.59	Does not satisfy criteria of brain kinetics	N.T.	N.T.	N.T.	High affinity for α-syn	Nonspecific binding to myelin sheaths
[^18^F]SQ3	39.3	230	N.T.	Yes, moderate selectivity	20	0.00426	Moderate brain permeability	2.08	Slow clearance rate	N.T.	N.T.	N.T.	Favorable pharmacokinetics in terms of brain permeability and stability against defluorination	Good binding affinity for α-syn aggregates and moderate selectivity
[^125^I]BI-2	99.5 ± 20.8	727 ± 227	N.T.	No	52	81.4	Passing BBB, low initial brain uptake	0.56	Slow clearance	N.T.	N.T.	N.T.	High selective binding affinity for α-syn aggregates	Low brain uptake and clearance
[^125^I]INPBF-3	0.28 ± 0.17	1.2 ± 0.55	N.T.	Yes, ∼4.4-fold	19.6	81,400	Passing BBB, low brain uptake, SUV < 1	N.T.	N.T.	N.T.	N.T.	N.T.	High affinity for α-syn	High logP value = 6.17
[^125^I]51	17.4 ± 5.6	73	N.T.	Yes, moderate selectivity	55.8	N.T.	Passing BBB, moderate brain uptake	3.57 ± 0.28	Good washout rate	N.T.	N.T.	N.T.	High affinity and good selectivity	High lipophilicity (cLogP = 5.15)
[^18^F]asyn-44	1.85 ± 0.38	170 ± 60	4600, or >10,000	Yes	6 ± 2	263 ± 121	Good brain permeability, SUV > 1.5	N.T.	Moderate washout	N.T.	N.T.	High binding to MSA and PD donors, weak binding in AD, no binding with tau	Favorable in vitro and in vivo characteristics for neuroimaging	Presence of radiometabolites in rat brain
[^11^C]APT-13	27.8 ± 9.7	92.6 ± 48.8	N.T.	Yes	13.5 ± 1.8	98.7 ± 12.7	Excellent brain penetration; SUV = 1.94 ± 0.29	N.T.	Fast washout (t_1/2_ = 9 ± 1 min)	N.T.	N.T.	-	Highest affinity for α-syn with good selectivity and favorable pharmacokinetic properties	Lack of studies using brain tissues and rodent models
[^125^I]ITA-3	>1000	3.7 ± 1.3	N.T.	No	16–75	N.T.	Satisfactory BBB permeation	4.9 ± 0.9	N.T.	N.T.	N.T.	-	Moderate affinity for α-syn (IC_50_ = 55 nM) in human PD brain sections	Slow clearance, high logP values
[^18^F]FITA-2	>1000	106.6 ± 9.8	N.T.	No	>25	>110	Passing BBB, good brain uptake, SUV_peak_ = 2.80 ± 0.45	5.4 ± 0.6	Fast clearance rate	N.T.	N.T.	High specific binding to α-syn pathologies in postmortem PD brain tissues	Moderate binding affinity to α-syn pathologies (IC_50_ = 245 nM)	Binding affinity to α-syn pathologies and selectivity was not optimal
[^18^F]F0502B	10.97	109.2	120.5	Yes	~10	74	High BBB permeability	N.T.	Rapid clearance	8 rhesus macaques, injection of PBS (2), AAVs encoding A53T mutated human α-syn (3) and α-syn PFFs (3), respectively	N.T.	High specific binding to α-syn pathologies	High binding affinity and selectivity to α-syn	Imaging characteristics in patients still need further investigation

RCY: radiochemical yield; AM: molar activity; PET: positron emission tomography; α-syn: alpha-synuclein; Aβ: β-amyloid; BBB: blood–brain barrier; SUV: standardized uptake value; PD: Parkinson’s disease; AD: Alzheimer disease; MSA: multiple system atrophy; PBS: phosphate-buffered saline; N.T.: not tested.

**Table 3 cells-14-00907-t003:** Summary table of α-syn radioligands identified through high-throughput screening.

Radiolabeled Compounds	In Vitro Assays				Preclinical Trial					Clinical Trial			Advantages	Limitations
	K_i_ or K_d_ (α-Syn Fibrils) (nM)	K_i_ or K_d_ (Aβ Fibrils) (nM)	K_i_ or K_d_ (tau Fibrils) (nM)	High Selectivity (α-Syn/Aβ)	RCY [%]	AM [GBq/μmol]	BBB Permeability with Early Peak Uptake of SUV	≥0.4% ID/g (Rat Brain) or ≥4.0% ID/g (Mouse Brain) [%]	Clearance Rate	PET Imaging in Animals	PET Imaging in Human Subjects	Tested In Vitro Autoradiography (Gold Standard)		
[^125^I]61	1.06	4.56	N.T.	Yes, ∼5-fold	57	81	N.T.	N.T.	N.T.	N.T.	N.T.	[^125^I]61 binds to sarkosyl-insoluble fraction in A53T mouse brain	High affinity with α-syn fibrils	Relatively high nonspecific binding
[^125^I]21	0.48 ± 0.08	2.47 ± 1.30	N.T.	Yes, ∼5.2-fold, suboptimal selectivity	12 ± 2	344 ± 235	Passing BBB with peak SUV of ~2.3	N.T.	Fast washout	Rhesus macaques (2)	PD (1); AD (1); HC (1); homogenate	N.T.	Rapid metabolic rate	High nonspecific binding
[^18^F]2FBox	3.3 ± 2.8	145.3 ± 114.5	N.T.	Yes, ∼44-fold	10–19	68–543	Good BBB permeation with peak SUV of 1.6	0.47	Moderate washout kinetics	N.T.	PD (1); MSA (1); HC (1); AD (1)	Cannot image Lewy bodies or neurites	High affinity with α-syn fibrils	No binding to α-syn aggregates confirmed by postmortem tissue investigation
[^18^F]4FBox	155.4 ± 96.5	7.7 ± 2.6	N.T.	Yes, ∼20-fold	10–19	68–543	Good BBB permeation with peak SUV of 1.6	0.47	Moderate washout kinetics	N.T.	PD (1); MSA (1); HC (1); AD (1)	Cannot image Lewy bodies or neurites	High affinity with α-syn fibrils	No binding to α-syn aggregates confirmed by postmortem tissue investigation
[^18^F]d_8_	0.1	386.3	>1000	Yes, 1/3863	≥25	40–104	Good BBB permeation	>4	Fast washout	N.T.	N.T.	N.T.	High binding affinity to α-syn; low lipophilicity	-
[^18^F]anle138b	190 ± 120	N.T.	N.T.	N.T.	N.T.	N.T.	Good BBB permeation	N.T.	N.T.	N.T.	N.T.	N.T.	-	High lipophilicity (logP = 4.34)
[^11^C]anle253b	N.T.	N.T.	N.T.	N.T.	47	15.1 ± 3.4	Clear penetration	0.25–0.3	Slow washout	N.T.	N.T.	N.T.	Direct binding to α-syn fibrils (IC_50_ = 1.6 nM)	High lipophilicity (logP = 5.21), lack of information about affinity using human-derived brain tissue
[^11^C]MODAG-001	0.6 ± 0.1	20 ± 10	19 ± 6.4	Yes, ∼30-fold	3.6 ± 1.1	98.6 ± 24.7	Good BBB permeation, SUV = 1.4	N.T.	Fast washout	N.T.	LBD (2); AD (1); PSP (1); HC (1)	Negative binding in in vitro ARG using DLB brain sections	High binding affinity to α-syn, no binding to tau in PSP and AD brain tissues	Detection of radiometabolites in mouse brain, high nonspecific binding in human LBD brain tissue
(d3)-[^11^C]MODAG-001	0.6	N.T.	N.T.	Yes	N.T.	N.T.	Passing BBB with rapid brain uptake, SUV = 1.7	N.T.	Fast washout	Female domestic pigs (4)	N.T.	N.T.	High binding affinity to α-syn fibrils and succeeded in detecting α-syn in fibril-inoculated rat model	-
[^11^C]4i	3.0 ± 1.4	N.T.	N.T.	Yes	8.0 ± 2.9	106 ± 56	Good brain permeability	1.68 ± 0.54	Rapid washout	Male rhesus macaque (1)	N.T.	Binding to α-syn in MSA and not PD and controls; binding affinity in PD tissue sections by ARG not as high as in vitro binding assays conducted in PD brain homogenates	Nanomolar binding affinity for α-syn, lower binding affinity for Aβ; high signal-to-background ratio	High affinity to aggregated tau proteins

RCY: radiochemical yield; AM: molar activity; PET: positron emission tomography; α-syn: alpha-synuclein; Aβ: β-amyloid; BBB: blood–brain barrier; SUV: standardized uptake value; HC: healthy control; PD: Parkinson’s disease; DLB: dementia with Lewy bodies; LBD: Lewy body disorders; AD: Alzheimer disease; PSP: progressive supranuclear palsy; MSA: multiple system atrophy; ARG: autoradiography; N.T.: not tested.

## Data Availability

The original contributions presented in this study are included in the article. Further inquiries can be directed to the corresponding authors.

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
