# Peer review of "Development of Positron Emission Tomography Radiotracers for Imaging α-Synuclein Aggregates"

_cells, 2025, doi:10.3390/cells14120907_

Round 1

Reviewer 1 Report

Comments and Suggestions for Authors

Re: cells-3589174

This is a narrative review on the development of radiotracers for PET imaging of α-synuclein and related disorders. Although radiotracers for PET imaging of Aβ amyloid and tau for Alzheimer’s disease are already in use in clinical practice, radiotracers for α-synuclein are not currently available in clinical practice, although urgently needed. Thus, the present review is welcomed, presenting the most recent advances in the development of such radiotracers.

The review is well written. Presented are Alzheimer’s disease-related tracers, scaffolds and analogs, as well as major challenges and limitations of these tracers for preclinical research and clinical use. Tables are used to summarize data in a very attractive way. Finally, in the discussion the authors do not forget to suggest that a multimodal imaging approach and multimodal biomarker integration, combining PET, MRI and fluid biomarker data are the key in the diagnostic approach in every-day practice.

The paper is very comprehensive and educative and may prove useful to neurologists, imaging specialists and scientists involved in research and/or clinical approach of neurodegenerative disorders. I suggest to be published in the present form.

Author Response

Thank you very much for taking the time and your nice comments on our manuscript.

Reviewer 2 Report

Comments and Suggestions for Authors

This is an exhaustive review of experimental ligands that potentially could label a-synuclein aggregates in Lewy bodies and thus be utilized for non-invasive imaging of Lewy bodies in several related neurodegenerative diseases (NDDs). The authors present 3 large Tables of the compounds studied. In descriptive paragraphs, the authors make a mistake in describing the first ligand (PIB) when they refer to its "high affinity in vitro (for a-synuclein) (Kd = 10.07 nM) and lower affinity for Aβ1–42 fibrils (Kd = 0.71 nM)". This is reversed, as a lower Kd (equilibrium dissociation constant) designates higher affinity. They seem to correct this type of misstatement later on in the paper, but this is an obvious error on their part.

Almost all of their compounds presented in Tables 1-3 either have not been tested in humans with NDDs, and/or do not demonstrate selective binding to aggregated a-synuclein. However, there are a few promising leads presented, and one is hopeful that these leads will be followed up.

Overall, I did not find this paper to be helpful. What is needed are appropriate clinical imaging studies with a compound that works as a selective ligand, not a tabulation of ligands that did not work. There did not appear to be any organizing principles, and I do not feel this paper is helpful as a review article, even though the data it presents undoubtedly are accurate. 

Reviewer 3 Report

Comments and Suggestions for Authors

Guo et al provide a review on advances in Positron Emission Tomography Radiotracers for Imaging α-Synuclein Aggregates. I have the following comments regarding this work:

  1. The context of specific radiotracer imaging should be placed in the context of general overview on the PET neuroimaging of parkinsonisms - Ref.  - Accumulation of Tau Protein, Metabolism and Perfusion-Application and Efficacy of Positron Emission Tomography (PET) and Single Photon Emission Computed Tomography (SPECT) Imaging in the Examination of Progressive Supranuclear Palsy (PSP) and Corticobasal Syndrome (CBS). Front Neurol. 2019 Feb 14;10:101. doi: 10.3389/fneur.2019.00101. PMID: 30837933; PMCID: PMC6383629.
  2. Authors should provide and extended perspective on the issue of co-pathologies in neurodegenerative diseases
  3. The significance of the specific neuroimaging highlighted in this review should be analyzed in the context of mechanism impacting aggregation
  4. The limitations of the asssessed in the studies indicated in the literature should be evaluated in a more critical mannet eg the acknowledgement of the coexistence of different pathologies
  5. The issue of BBB peremeabiliry should be more higlighted in the context of alpha-syn aggregation

Round 2

Reviewer 2 Report

Comments and Suggestions for Authors

This is the first revision of a paper I originally reviewed and rejected. I need to be clear that I am not disputing the veracity of the radioligand data generated by other investigators who are appropriately referenced. Rather, I reviewed this paper as a Review Article; and I continue to feel that it is too preliminary for that genre.

The authors have corrected their referencing. Otherwise I can detect some changes in the text which improve clarity but do not alter substance.

I quote from the Conclusion section:

"Most of the compounds discussed here demonstrated nanomolar affinity for α-syn aggregates alongside BBB permeability and uniform brain distribution in preclinical models in vivo. Despite these advancements, no radiotracer has yet achieved clinical validation, with current candidates facing persistent challenges in distinguishing Aβ/tau heterotypic aggregates in co-pathological conditions-a critical limitation compounded by the high prevalence of mixed proteinopathies in neurodegenerative disorders."

Stated another way-none of the ligands presented were helpful in distinguishing mixed protein aggregates. Given that alpha-synuclein aggregation is itself NOT specific to any clinical condition (as discussed by the authors in their comprehensive Introduction), the authors appear to be cataloguing multiple failures to achieve protein aggregate binding specificity.

They continue:

"Emerging candidates, such as 18F-C05-05, 18F-SPAL-T-06, [18F]ACI-12589, (d3)-[11C]MODAG-001 and [18F]F0502B, exhibit subnanomolar affinity and optimized pharmacokinetics but require human validation (bolding mine) to confirm their diagnostic utility." 

Thus, according to this statement, none of the radioligands presented by the authors have been validated in human subjects (likely because their non-specificity precluded human subject approval for experimental testing). The authors have compiled large tables describing many attempts to develop such radioligands, but none have been tested in humans. What is to be gained by such an encyclopedic compilation of negative/preliminay data?

Again, I have no objection to the data presented. Rather, my concern is that this paper serves no meaningful purpose, other than to show that radioligands that bind with high affinity to alpha-synuclein exist, but that they are not specific to alpha-synuclein, which itself is not a disease-specific marker. I will leave the publication decision to the Editor.

Reviewer 3 Report

Comments and Suggestions for Authors

Authors did not implement the suggestion concerning putting alpha-synuclein radiotracer in the context of other radiotracers including tau-radiotracers.

Author Response

Comments: Authors did not implement the suggestion concerning putting alpha-synuclein radiotracer in the context of other radiotracers including tau-radiotracers.

Response: We sincerely thank the reviewers for the suggestions to combine the development of α-syn radiotracers with other radiotracers including tau-radiotracers. Although our review mainly focuses on the unique challenges and progress in the development of α-syn PET tracers, we admit that cross-comparison with tau tracers can enrich the discussion. In the section of "Challenges in developing α-syn PET tracers", we quoted the recent cross-pathology research review (Tracer development for PET imaging of proteinopathies, Annukka Kallinen, 2022) to further express the different characteristics of a-syn development with abeta/tau radiotracers. (Page 4, line 2)

Round 3

Reviewer 2 Report

Comments and Suggestions for Authors

This is my review of the second revision of a paper I have rejected twice. The authors have modified the Conclusion/Discussion section to indicate that human trials are needed for the best candidates, and that as of yet, there are no ideal ASN imaging candidates but several promising leads. Hopefully, their extensive database of ASN imaging radioligands will inspire additional research until molecular “tweaking” produces a useful reagent. One is surely needed, and the authors have gone to a great deal of trouble to compile a tabular history of many partial successes.

Reviewer 3 Report

Comments and Suggestions for Authors

The work was insufficiently improved. authors have not implemented my suggestions.

Response: Reviewer 3 suggested that the development of α-syn radiotracers should be combined with other radiotracers (including tau-radiotracers) and recommended one reference (PMID: 30837933; PMCID: PMC6383629). Although this manuscript focuses on the unique challenges and progress in the development of α-syn PET tracers, we admit that cross-comparison with tau tracers can enrich the discussion. Thus, in the section of "Challenges in developing α-syn PET tracers", we quoted the recent cross-pathology research review (Tracer development for PET imaging of proteinopathies, Annukka Kallinen, 2022) to further express the different characteristics of a-syn development with abeta/tau radiotracers. (Page 4, line 2). As for the addition of the reference on the perfusion and metabolism studies, the Academic Editor suggested that “it is not necessary to include a section on perfusion and metabolism studies as recommended by Reviewer 3 and this special issue of Cells is focused on imaging a-Syn”. Thus, we followed the Academic Editor’s suggestion and did not add the suggested reference.